depression; over-weight; body-image; weight-stigma; Palestine

**Corresponding author:**
Fayez Mahamid;
Email: Mahamid@najah.edu

# Differences in severity of depression symptoms in overweight, obese and normal weight Palestinian children and adolescents

Hadeel Agbaria[1], Fayez Mahamid[1] and Dana Bdier[2,3]

[1]Psychology and Counseling Department, An-Najah National University, Nablus, Palestine; [2]Psychology and Counseling Department An-Najah National University, Nablus, Palestine and [3]Department of Human Sciences and Education, R.Massa University of Milano-Bicocca, Milan, Italy

## Abstract

Obesity is related to a wide variety of medical and psychological comorbidities which has short- and long-term effects on children's mental health. One of the most significant ones is depression. Thus, the current study utilized a descriptive methodology to explore the differences in depressive symptoms among overweight, obese, and normal-weight Palestinian children and adolescents. Data was collected from 270 Palestinian children and adolescents, aged (9–16) years: 85 with normal weight, 95 with over-weight and 90 obese. Findings showed that participants who are over-weight or obese exhibited more depressive symptoms than those with a normal weight. These findings showed that Palestinian children and adolescents who are over-weight or obese do experience depression and thus interventions should take this into account. In particular, it seems that over-weight boys or adolescents need more direct help in losing weight while obese children and adolescents who feel more helpless about their weight need serious psychological interventions. it is critical to offer psychological treatment as part of any weight loss intervention program for children and adolescents. Especially as these adolescents' families might encourage them to avoid seeking professional help and deal with the problem in the family.

## Impact statement

Obesity is associated with numerous psychological and social issues in children and adolescents. One of the most serious consequences for overweight children is depression, which often stems from stigma and the social and familial pressures associated with excess weight, as well as the expectation to meet societal standards for an ideal body. Depressive disorders are exacerbated in overweight Palestinian children who face challenging circumstances, including political violence and occupation. This situation necessitates the development of therapeutic interventions aimed at enhancing the mental health of these children and adolescents, helping them develop a positive self-concept to cope with social pressures and reduce their depressive symptoms, as well as encouraging their physical activity in a positive way.

## Theoretical background

The prevalence of overweight and obesity almost doubled worldwide since the 1980s and continues to grow in all age groups and thus constitute a global problem (Skinner et al., 2015). These are conditions determined by the body mass index (BMI). BMI is a measure based on a calculation of a person's weight in relation to height taking into account the person's sex and age. In adults the World Health Organization (WHO) defines, a BMI of 25–29.9 as 'over-weight' and one that is over 30 as obesity. In children and adolescents, overweight is defined as a BMI of 85% or more in relation to age and sex and obesity is defined as 95% or more (Luppino et al., 2010).

Obesity is a result of an interaction of genetics and environmental factors, significant risk factors for obesity are having an obese parent (Wu, 2006), unhealthy family life style and low physical activity (Lee & Yoon, 2018). Among children and adolescents, obesity has been found to increase the risk of varied medical and mental health problems with possible long-term effects (Abdeen et al., 2012). Psychologically, obesity has been found as related to low self-esteem, anxiety, loneliness, low quality of life and depression (Kumar & Kelly, 2017). One of the main reasons for these symptoms is the prevalent weight stigma that blames obese children or adolescence for their over-weight and perceive them negatively in light of it (Palad et al., 2019). As a consequence, obese children and adolescence are often victims of teasing and harassment by family and peers and experience social exclusion (Haqq et al., 2021).

In Palestinian traditions, obesity is often viewed as a result of poor dietary management and an inability to regulate food intake in relation to activity levels. However, this perspective is overly simplistic, as obesity is a complex condition influenced by a combination of behavioral, environmental, genetic and biological factors (Haqq et al., 2021). Many interventions designed to prevent or treat childhood obesity in the Palestinian context reinforce existing stigmas by holding children responsible for altering their weight through changes in behavior. These initiatives often focus on personal responsibility rather than addressing broader factors (Harriger & Thompson, 2012). Beyond diet, a common belief in Palestinian culture attributes the rise in obesity over the past few decades to the sedentary lifestyles of many children and adolescents. Another prevalent belief is that criticizing and shaming individuals about their weight serves as motivation for them to lose it. However, such shaming can have serious psychological repercussions and may lead to unhealthy eating behaviors (Haqq et al., 2021). In Palestine, Overweight or obese children and adolescents often encounter stigma from their communities, family members, and peers, which can result in criticism, harassment, and unequal treatment. This stigma can lead to victimization, social inequality, marginalization, and negative health outcomes.

Specifically, obesity has been found to have a bi-directional relationship with depression among children and adolescents (De Wit et al., 2010; Incledon et al., 2011; Mannan et al., 2016; Mühlig et al., 2016). Depression symptoms in children or adolescents might include anger or hostility, changes in sleep or appetite, hopelessness, difficulty concentrating, lack of motivation and feelings of worthlessness (APA, 2013). Depression is more common among females and tends to get worse as children, especially females, get older. These differences are a result of the hormonal changes adolescents go through at this age that affects their level of stress and how they cope with it. Depression can significantly interfere with the adolescent's development, school achievement and relationships (Rao & Chen, 2009). It also increases their risk of death as a result of suicide. Older adolescents, 15–19 years old were found to conduct one of the highest number suicides in the U.S. in 2021 (WHO, 2021).

Obesity was found to increase levels of depression, through victimization, social rejection, low self-esteem and loneliness (Haqq et al., 2021). In addition, depression increases the risk of obesity. The reasons for that are the depressive symptoms might increase appetite and sedentary behavior and thus result in gaining weight (Mannan et al., 2016). Studies examining this issue found inconsistent effects of gender and age on the relationship between obesity and depression. Some studies found that these relationships are stronger among females (Korczak et al., 2013; Mühlig et al., 2016; Lindberg et al., 2020; Wyshak, 2007). However, in other studies, no differences between genders were found in the relationship between obesity and depression (Kubzansky et al., 2012; Lindberg et al., 2020; Mannan et al., 2016). Regarding the effect of age, some studies conducted with obese children aged 6–18, found that their depressive symptoms became more severe as they got older (Lindberg et al., 2020). However, in a longitudinal study, no interaction between age and weight was found for level of depression (Kubzansky et al, 2012).

Studies examining the association between obesity and depression have looked at both directions of this relationship. On the one hand, it was found that the impaired sleeping and eating that is part of depression, tends to increase depressed people's adoption of unhealthy lifestyle. They prefer eating high carbohydrate, high sugar, comfort foods and avoid physical activity. These behaviors place them at increased risk of weight gain and obesity (Privitera et al., 2013). In parallel, several studies found that people who are obese, are at a higher risk to develop depression, compared to those with normal weight (Carpenter et al., 2000; Fleiti, 1993). Roberts et al. (2000), also showed that obesity predicts future (one year later) depression.

Historically, most studies on the relationship between obesity and depression were conducted with adults. The focus of research on children and adolescents, in this context, has been growing due to the increase in the prevalence of obesity in youth. Similar to adults, studies show there is an association between obesity and depression among children and adolescents. In a meta-analysis of studies examining this issue, more than half of the cross-sectional studies (five out of eight) found significant correlations between obesity or over-weight and depression in this age group (Mühlig et al., 2016). In addition, in a study that compared between high school students of normal weight, over-weight and obese – BMI was correlated with severity of depressive symptoms (Kubzansky et al., 2012). The relationships between obesity and depression in children and adolescents is also apparent in similarities in some of the clinical presentations of these conditions. First, both affect sleep. Overweight and obese children are at higher risk for sleep apnea and decreased nighttime sleep (Barlow & Diez, 1998; Gupta et al., 2002). Children and adolescents with depression, also suffer more from insomnia and difficulty falling asleep (Emslie et al., 2001). Second, they both affect the person's self-image. Depression leads to feelings of guilt and shame and contributes to negative self-esteem (Weiss et al., 1992). Obesity often leads to body image issues, particularly among female adolescents (Franklin et al., 2006; Witherspoon et al., 2013).

This relationship was found to be bi-directional. On the one hand, childhood or adolescent obesity were found to predict subsequent depression (De Wit et al., 2010; Goldfield et al., 2010; Goodman, & Whitaker, 2002; Mühlig et al., 2016; Stunkard et al., 2003; Swallen et al., 2005). In a meta-analysis of 13 studies, adolescents who are obese, had a 40% higher risk to be depressed at a second measurement (Mannan et al., 2016) and as adults, than those who are not (Anderson et al., 2007; Herva et al., 2006). One of the reasons offered for this direction of the relationship, is the stigmatization of obesity (Goldfield et al., 2010; Puhl & Heuer, 2009).

In parallel, adolescents with symptoms of depression, were found to gain weight dramatically through a year of study (Incledon et al., 2011). In a meta-analysis of 13 studies, adolescents who were depressed, had a 70% higher risk to be obese at the second measurement, compared to those who were not (Mannan et al., 2016). In addition, depressive symptoms during childhood or adolescence increase the risk of obesity in adulthood (Blaine, 2008; Franko et al., 2005; Goodman & Whitaker, 2002). As found for adults, children and adolescents who are depressed, tend to be less active, adopt more sedentary behaviors and increase eating unhealthy comfort foods and eating in general. All this might increase their weight (Reeves et al., 2008). In addition, emotional distress often led to increased food consumption, especially comfort food, including high fats and sugars and irregular meal times, which might result in weight gain.

Children/adolescents experiencing emotional distress might use food and over-eating as a maladaptive coping mechanism to suppress negative emotions and avoid dealing with them. This increases their risk of obesity (Incelton et al., 2011; Lee & Yoon, 2018).

In addition to studies investigating one direction of this relationship, other studies looked at factors that might affect both obesity and depression. For example, one explanation offered was that obesity is a pro inflammatory state (Ferrante, 2007) and thus increase the risk of depression which is perceived as a dysregulation of the inflammatory system (Kim et al., 2007; Nemiary et al., 2012).

In some studies, the relationships between over-weight and depression were affected by gender and age. For example, in a meta-analysis of cross-sectional studies, it was found that these relationships are stronger among females (Mühlig et al., 2016). Similarly, in a study of obese children ages 6–18, depression was more common among girls than among boys (Lindberg et al., 2020). In addition, female obese adolescents had a higher risk to develop depression in adulthood than normal weight ones (Korczak et al., 2013). It was also found that girls exhibited severe depressive symptoms when their BMI was high, while for boys, the opposite was found, their depression was severe, when their BMI was normal (Carpenter et al., 2000; Heo et al., 2006). In parallel, studies showed that women with childhood depression are at higher risk to develop over-weight and obesity as young adults, compared to those not depressed (Korczak et al., 2013; Wyshak, 2007). However, in other studies, no differences between genders were found in the relationship between obesity and depression (Kubzansky et al., 2012; Lindberg et al., 2020; Mannan et al., 2016).

Another demographic factor that was found to affect the relationship between obesity and depression in children and adolescents, though not consistently, is age. In a study that was conducted with obese children aged 6–18, their depressive symptoms were more severe, the older they got (Lindberg et al., 2020). Similar findings were found in a meta-analysis of cross-sectional studies (Mühlig et al., 2016). However, in a longitudinal study that followed high school students with different weights (normal weight, over-weight, obese) for five years, it was found that the relationship between their weight and distress was stable over the years. In other words, even though the heavier weight and obese groups exhibited severe depressive symptoms, the relative intensity of their distress, compared to youth with lower weight, did not change the older they grew (Kubzansky et al., 2012).

### The current study

As presented above, studies conducted in Western cultures have demonstrated a bi-directional relationship between obesity and depression (Mannan et al., 2016; Mühlig et al., 2016). Explanations offered for it focused on one direction of the relationship or attempted to find factors that influenced both. In spite of this vast body of literature, these conditions are still treated separately as the understanding of the risk factors for the combination of them is still lacking or inconsistent. For example, important elements critical in identifying children or adolescents who are in particular at risk to develop this combination are demographic characteristics as sex and age. However, the findings regarding their effects on the strength of the relationship between obesity and depression is inconsistent. Thus, it is important to study this question further to be able to identify those at the highest risk. In addition, most of the research done in this issue up to now has been conducted in Western individualistic societies and is thus less applicable to traditional eastern cultures. As traditional collectivistic cultures perceive gender roles, social norms and mental illness differently (Daradas et al., 2016), it stands to

reason that depression in general and the relationship of obesity to it will be manifested differently in such a society. In particular, as the status of girls is lower in such societies, it is important to examine whether among adolescent girls in collectivistic societies, such as the Palestinian society, who exhibit over-weight or obesity suffer more from depressive symptoms. Another question that was left unresolved in the literature and has been tested in the current study is whether and to what degree is this relationship affected by age. This again will allow better identification of adolescents who need emotional support and tailor any intervention provided to them to take cultural issues into account. Furthermore, as Palestinians are exposed to war and political conflict, Palestinian children and adolescents are found to experience personal trauma or witness trauma to others due to occupation (El-Khodary et al., 2020). Experiencing such traumatic events has been found to correlate positively with depression among Palestinian children and adolescents. According to a scoping review that explored the consequences of war-related trauma reactions among Palestinian children and young people in the Gaza Strip, depression was one of the most prevalent mental health outcomes (Abudayya et al., 2023). Additionally, the results of a review paper revealed that the prevalence of depression ranged from 40% in children of Gaza and the West Bank to 50.6% as a result of exposure to political traumatic events due to occupation (Thabet, 2019).

Therefore, the current study hypothesized. First, Palestinian children and adolescents who are obese will have severe symptoms of depression than those who have normal weight. Second, Gender affects the relationship between obesity and symptoms of depression. Over-weight and Obese girls will have severe symptoms of depression than over-weight and obese boys. Third, age affects the relationship between obesity and symptoms of depression. Over-weight and obese adolescents will have severe symptoms of depression than over-weight and obese children.

## Methods

### Sample and procedures

The sample included 270 Palestinian Muslim participants aged 9–16 (M = 13.07, SD = 2.07) years; 50.6% of participants were females, and 49.4% of participants were males. 50% of participants were of normal weight, 35.7% of participants were overweight and 14.3% were obese. After receiveing the ethics approval by An-Najah Instituationl Review Board (IRB) in November 2023, the principles of the 10 Palestinian governmental schools (five schools are located in the city of Umm al-Fahm, and five schools in Kafr Qara) chosen for the study were approached. After receiving their approval to conduct the study with their students, school nurses and teachers helped identify students that fit the weight criterions in the 10 groups (normal weight, over-weight and obese) required for the study according to their medical diagnoses kept in their school nurse's medical file. Children and adolescents recruited to the study and their parents were provided with a detailed explanation of the type and purpose of the study, the duration of the session and that all the informaiton collected will be kept confidential. They were assured that the children would not be harmed in any way and that they will be able to leave the study at any point without a problem if they feel uncomfortable. Both the consent of the children and the parents were received. We sorted the students according to the CDC curves of child BMI to three groups (normal weight, over-weight and obese).

## Measures

Following standard methodological recommendations for questionnaire development (Hambleton et al., 2005), all measures not already validated in Arabic were translated and back-translated from the original English version into Arabic. This process involved a panel of 10 Arab professionals in psychology, counseling and social work who evaluated the clarity and relevance of the questions and translations. After completing the initial draft of translated items, the questionnaires were back-translated into English by an independent expert English editor. Based on their feedback, the translated version was pilot-tested among 80 participants and further refined for clarity.

### Demographic variables

The study included the following demographic variables: age, gender, school and weight status (classified as normal weight, overweight and obese).

### Depression symptoms

*The Children's Depression Inventory (CDI2), (kovacs, 1992):* In order to measure the severity of the participants' Depression symptoms, the CDI2 was used. This scale is intended for children between the ages of 7 and 17. It contains 28 items, measuring four types of Depression symptoms: negative self-esteem (6 items), inactivity (8 items), difficulties in interpersonal relationships (5 items) and symptoms of negative mood and body (9 items) (for example: "Feeling sad". On each item, participants were asked the frequency they experienced what is described in the item on a three point lickert scale (all the time, most of the time and rarely). Five scores were calculated for each particpant for this scale – for each sub scale and a total score by summing the rating they provided.

## Data analysis

In order to test the research hypotheses, descriptive statistics were calculated for depressive symptoms total score, difficulties in interpersonal relationships, inactivity, negative self esteem and mood and body image. Moreover, a one-way MANOVA was performed as the most suitable statistical method to identify differences among independent groups (normal weight, overweight and obese) in relation to dependent variables, including interpersonal relationship difficulties, inactivity, negative self-esteem, mood and body image and total depression symptoms. Additionally, a two-way MANOVA was utilized to investigate the differences in depressive symptoms across age and gender within the overweight and obese categories. Finally, Tukey's HSD tests was used to interpret the

statistical significance of the difference between means of the three groups (normal weight, overweight and obesity) on the depression scale.

## Results

As can be seen in Table 1, participants exhibited relatively low levels of total depression score. They also reported low scores on difficulties in interpersonal relationships, inactivity, negative self esteem, and mood and body image sub-scales.

As can be seen in Table 2, in each of the depression symptoms, significant differences were found between at least two groups: depression symptoms total ($F$ 2,87 = 54.17; $p < 0.01$), difficulties in interpersonal relationships ($F$ 2,87 = 19.31; $p < 0.001$), inactivity symptoms ($F$ 2,87) =26.40; $p < 0.01$), negative self esteem ($F$ 2,87 = 20.6; $p < 0.01$), mood and body image ($F$ 2, 87 = 26.85; $p < 0.01$). In order to idenfidy the groups that are significantly different in each depression measure Post hoc Tukey tests were conducted comparing between each two groups in each measure. The Tukey's HSD tests showed that: the mean value of total depression symptoms in the normal weight group (M = 7.8, SD = 4.6) were significantly different at ($p < 0.001$) than in the overweight group (M = 20.07, SD = 4.46) and from those in the obesity group (M = 21.77, SD = 7.45). However, the differences between the total depression symptoms in the overweight and obese groups were not significantly different. Similarly, the Tukey's HSD tests showed that the mean value of difficulties in interpersonal relationships in the normal weight group (M = 1.33, SD = 1.34) were significantly different at ($p < 0.001$) than in the overweight group (M = 3.53, SD = 1.25) and from those in the obesity group (M = 4.0, SD = 2.46). However, the differences between difficulties in interpersonal relationships in the overweight and obese groups were not significantly different. Similarly, the Tukey's HSD tests showed that the mean value of inactivity in the normal weight group (M = 2.43, SD = 1.97) were significantly

**Table 1.** Descriptive statistics for depression scores

|  | M | SD | Min | Max |
|---|---|---|---|---|
| Depression symptoms total | 16.54 | 8.4 | 0 | 37 |
| Difficulties in interpersonal relationships | 2.96 | 2.19 | 0 | 13 |
| Inactivity | 4.73 | 2.67 | 0 | 11 |
| Negative self esteem | 2.78 | 2.02 | 0 | 7 |
| Mood and body image | 6.08 | 3.41 | 0 | 14 |

**Table 2.** Comparision of depressive symptoms between weight groups

|  | Normal weight ($n$ = 135) | | Overweight ($n$ = 96) | | Obesity ($n$ = 39) | | | |
|---|---|---|---|---|---|---|---|---|
|  | M | SD | M | SD | M | SD | $F$ | $\eta2$ |
| Depression symptoms total | 7.8 | 4.6 | 20.07 | 4.46 | 21.77 | 7.45 | ***54.17 | 0.52 |
| Difficulties in interpersonal relationships | 1.33 | 1.34 | 3.53 | 1.25 | 4.0 | 2.46 | ***19.31 | 0.27 |
| Inactivity | 2.43 | 1.97 | 5.7 | 1.95 | 6.06 | 2.43 | ***26.4 | 0.36 |
| Negative self esteem | 1.16 | 0.95 | 3.56 | 1.99 | 3.6 | 1.9 | ***20.6 | 0.29 |
| Mood and body image | 2.86 | 2.17 | 7.26 | 2.13 | 8.1 | 3.16 | ***26.85 | 0.34 |

***$p < 0.001$.

**Table 3.** Comparing depressive symptoms between genders in over-weight and obesity groups

| | Overweight | | Obese | |
|---|---|---|---|---|
| | Female (*n* = 42) | Male (*n* = 54) | Female (*n* = 13) | Male (*n* = 26) |
| Depression symptoms total | 20.93 (4.2) | 19.31 (4.67) | 24.69 (6.56) | 19.53 (7.48) |
| Difficulties in interpersonal relationships | 3.79 (1.12) | 3.31 (1.35) | 4 (2.46) | 4.31 (1.6) |
| Inactivity | 6.21 (2) | 5.25 (1.84) | 7.07 (2.78) | 5.29 (1.86) |
| Negative self esteem | 4.14 (2.25) | 3.06 (1.65) | 4.08 (1.66) | 3.24 (2.05) |
| Mood and body image | 6.79 (1.81) | 7.69 (2.36) | 9.23 (2.49) | 7.24 (3.42) |

**Table 4.** Comparing depressive symptoms between ages in over-weight and obesity groups

| | Overweight | | Obese | |
|---|---|---|---|---|
| | Child (*n* = 47) | Adolescent (*n* = 49) | Child (*n* = 18) | Adolescent (*n* = 21) |
| Depression symptoms total | 18.27 (4.1) | 21.11 (8.55) | 23.58 (5.9) | 20.56 (8.26) |
| Difficulties in interpersonal relationships | 3.36 (1.36) | 3.63 (2.08) | 3.92 (1.31) | 4.06 (3.04) |
| Inactivity | 5.45 (1.69) | 5.84 (2.67) | 7.08 (2.5) | 5.39 (2.2) |
| Negative self esteem | 3.18 (2.23) | 3.79 (1.99) | 4.17 (1.47) | 3.22 (2.1) |
| Mood and body image | 6.27 (2.24) | 7.84 (3.41) | 8.42 (2.35) | 7.89 (1.66) |

different at ($p < 0.001$) than in the overweight group (M = 5.7, SD = 1.95) and from those in the obesity group (M = 6.06, SD = 2.43). However, the differences between inactivity in the overweight and obese groups were not significantly different. Similarly, the Tukey's HSD tests showed that the mean value of negative self esteem in the normal weight group (M = 1.16, SD = 0.95) were significantly different at ($p < 0.001$) than in the overweight group (M = 3.56, SD = 1.99) and from those in the obesity group (M = 3.6, SD = 1.9). However, the differences between the negative self esteem in the overweight and obese groups were not significantly different. Similarly, the Tukey's HSD tests showed that the mean value of mood and body image symptoms in the normal weight group (M = 2.86, SD = 2.17) were significantly different at ($p < 0.001$) than in the overweight group (M = 7.26, SD = 2.13) and from those in the obesity group (M = 8.1, SD = 3.16). However, the differences between mood and body image symptoms in the overweight and obese groups were not significantly different. In summary, overweight and obese Palestine children and adolescents exhibited higher depressive symptoms of all types than normal weight ones.

The MANOVA analysis(see Table 3) for the total depression symptoms, inactivity and negative self esteem found an effect of gender (depression total $F(1,56) = 4.86$, $p = 0.03$; inactivity $F(1,56) = 6.22$, $p = 0.02$; negative self esteem $F(1,56) = 3.71$, $p = 0.06$ (marginal signficance), manifesting higher levels of depression among females than males, the MANOVA analysis for mood and body image symptoms found no category or gender effects but found a significant interaction between category and gender ($F(1,56) = 4.5$, $p = 0.04$).

The MANOVA analysis (see Table 4) for total depression symptoms found no category or age effects but found a marginally significant interaction between category and age group ($F(1,56) = 3.3$, $p = 0.075$). Among overweight participants, adolescents exhibited worse total depression symptoms than children, while the opposite trend was found for obese participants.

## Discussion

The aim of the current study was to investigate the relationship between weight and depression among Palestinian children and adolescents. Our findings showed that the depression symptoms (of all types) of over-weight and obese Palestinian children and adolescents were higher than that of normal weight children/adolescents. These findings are similar to those found in previous studies (Stunkard et al., 2003). As the current study included only one point of data collection, there is no definitive way to conclude the direction of this relationship. In other words, according to the literature it is possible that over-weight or obesity are a result of the depression (Privitera et al., 2013) or vice versa (Roberts et al., 2000).

It is important to note that, in contrast to our hypothesis, no differences were found between the over-weight and obesity group participants in the severity of any of the depression symptoms. This finding might indicate a lack of direct relationship between weight and depression among Palestinian children and adolescents. In other words, it seems that the effect of over-weight on depression occurs no matter what the extent of the over-weight is and that there is no relation between this extent and the severity of their depression. A possible explaination for this finding is a methodological limitation that resulted in similarities in the weights of some of the participants in the two over-weight groups. Among participants, most of those defined as over-weight, had a bmi that was very close to the 97 percentile defined as obese. These small weight differences might have made it harder to identify differences between the over-weight and obese participants in level of depression. Aside from the methodological explanation, it is possible that children/adolescents are especially sensitive at this age to any body changes and thus it matters less the extent of the over-weight, only its existence. As previous studies showed, the main reason found for the effect of over-weight or obesity on depression was through lowering self-esteem and stigmitization Goldfield et al., 2010). For this effect the extent of over-weight might be less significant. It should be stressed that as this study was conducted with a different population, a collectivistic cultture, it might also be that this finding is specific to this type of culture. In other words, it is possible that in these cultures, any difference from others is apparent and cause for concern as it harms the family or group harmony and it does not matter the extent of the difference. In order to check the specificity of this finding to this type of culture, future studies in other populations are needed.

Similar to most previous studies done globally and in Arab countries (Daradas et al., 2016; Raccine et al., 2021), Palestinian girls exhibited severe depression symptoms than boys. This was manifested in total depression, inactivity symptoms and negative self esteem. These findings support previous studies and might be a

result of the gender norms in the Arab society that increase the risk of girls to develop depression (Daradas et al., 2016).

Regarding the effect of gender on the relationship of weight with depression, it was found that it only effected its relationship with mood and body image symptom. Among the obese participants, females exhibited worse mood and body image symptoms than males. Among the overweight participants, a small oppositie trend was found where males exhibited more depression symptoms than females. It seeems that this pattern of finding parallel that of previous studies among females severity of depression is related to high BMI, while among males it is related to normal or close to normal BMI (Heo et al., 2006). However, when one looks at the BMI distribution of the participants, one can notice that the average BMI of most males in the over-weight group, across age groups, were close to the 97th precentile signifying obesity. As such, it seems, that even males defined in this study as over-weight had a weight that was very close to being defined as obese. Thus the interaction finding actually shows that in both genders, their body image is affected by being very over-weight, and this effect is more extreme for females. This finding is supported by that of a previous study showing a strong relationship, in both genders, between obesity in adolescents and low body image (Gouveia et al., 2014). Our findings show that for Palestinian adolescents as well, both boys and girls are sensitive to stigmitization around their weight and thus it affects their body image similarly.

The second demographic variable tested in relation to weight and depression is age. In general, age and depression symptom severity were not found as related in any of the weight groups, and symptoms. In addition, depression symptoms severity was not significantly different between children and adeolscents in any of the depression symptoms. However, in support of our hypothses, two marginally significant interaction effects were found between age and weight groups, for total depression score and for inactivity symptoms. In both cases, among overweight participants, adolescents exhibited more total depression symptoms than children, while the opposite was found for obese participants. These findings appear to present an opposite trend to previous findings (Lindberg et al., 2015; Mühlig et al., 2016) in that the relationship between weight and depression is stronger for younger children and not for older.

One possible explanation for these findings is that over-weight and obseity are perceived differently by children/adolescents. It might be that overweight is seen as a state that is easier to fix or correct by a change of life style than obesity which requires more extensive and encompassing treatment. As such, when it comes to over-weight, adoelscents might believe that they should be able to reduce their weight on their own (or with some guidance) and if they cannot it is their own failure. Children are probably less susceptable to this bias as they might tend to rely more on their parents to decide on their diet and any changes to it. As a result, over-weight adolescents might feel more depressed than children.

However, when it comes to obesity, which is perceived more as a situation they cannot cope with on their own – they are helpless about – a different process will come into play. In this case, children might present worse depression as they will suffer worse stigmitization and be more affected socially by the way they look. This might be especially significant in the current study where the children were relatively older 9–12, early adolescence, when they develop their sense of self and body image (Marmorstein et al., 2014). As such, obesity might have a particularly strong affect on level of depression.

## Limitations of the study

Our study has a few limitations that may offer opportunities for future studies. First, the study's sample was not representative and included only Palestinian Muslim children and adolescents aged 9–16 years. Thus the findings could not be generalized to the population of Palestine or the Arab world. In addition, the children in each weight group were no sampled randomly but were provided for the study by the schools where the study took place. As such, they do not necessarily represent Palestinian children. Thus in the future it is important to sample from wider groups of children and choose among them those in the different weight groups.

Second, the study included collection of data only at one point in time. This prevented us from being able to make conclusions regarding the direction of the relationship between obesity and depression and examine changes in each of these over time and their mutual effects. It is important in a future research to conduct a longitudinal study with at least two data points that will allow answering these questions. Third, the tools used in this study, the depression scale, was not validated in the Arab language as there is no Arabic depression scale. This is the first time a translation of the scale was used and examined. Future studies are needed for further testing of its reliability and validity. Fourth, the study was conducted only in one culture group – collectivistic. It did not conduct direct comparison between adolescents in collectivistic and individualistic cultures and thus the research conclusions might be limited. Future studies need to conduct direct comparisons between the two. Finally, we targeted Palestinian children in the West Bank of Palestine during a difficult period characterized by many conflicts and political violence between Palestinians and Israelis. Hence, the political violence heightened feelings of depression among Palestinian children, possibly skewing the findings. More future studies are recommended to test the correlation between current study variables over different periods of time.

## Conclusions

In summary, the study findings showed that similarly to other western and Arab countries, Palestinian children and adolescents who are over-weight or obese are more depressed than those who have a normal weight. The symptoms of depression among Palestinian children and adolescents are exacerbated by the difficult conditions faced by the Palestinian community, such as occupation and ongoing trauma. These children are continuously exposed to arrests, house demolitions, and daily incursions into towns and villages, which increases their psychological suffering. Children who struggle with obesity face additional challenges in these circumstances, as psychological and social pressures impact their mental health even more. They may experience social stigma or rejection from their peers, which reinforces their feelings of isolation. As this study is not longitudinal, there is no way of knowing for sure the direction of this relationship but in light of the effect found for having an obese family member, it seems likely that over-weight or obesity is the one effecting the children/adolescents' depression. In other words, that at least some Palestinian families adopt unhealthy eating habits or are more accepting to being over-weight, which increase the risk of their children to be over-weight and obese. Being over-weight or obese puts these youth at risk for developing internalizing disorders like depression as a result of stigmitization. Similar findings were found for the two the over-weight and obesity groups, which might show that at least in this population the extent of the over-weight is less relavent to the

severity of their depression. Just them being different than others is what results in depression.

Regarding the effect of gender, we found similar relationships between weight and depression in both boys and girls. In other words, young and old adolescent males who are over-weight or obese also experience depression and not just the girls. In particular, when overweight males are unsatisfied with their weight and maybe feel they should be able to lose it, their depression levels are high. This group of males experiences depression out of a sense of personal failure of not being able to lose weight and feel their being over-weight is on some level their "own fault". Their body represents for them their inability to change how they look and thus their body image is worse than over-weight females. Obese females exhibited negative body image than obese males. This group of participants were also not satisfied with their weight but they felt helpless to change it. As they gave up the idea of losing weight they were not physically active at all.

Regarding age differences, this study showed different patterns among the over-weight and obese participants. For the over-weight participants, it was found that older adolescents exhibited more depressive symptosm than young ones. However, for obese youth, the opposite pattern was found. This pattern is different than the one found in many previous studies that had showed that the relationship between weight and depression increased with age. Similar to the findings regarding gender, these findings help us understand more deeply how depression symptoms are developed differently in young and old adolescents when it comes to being over-weight. Among over-weight participants, older adolescents experience more depressive symptoms than younger ones. This group wish they could lose weight and feel it is their own fault they are not thinner. The younger adolescents have less control on their diet and life style and thus take less self-blame for their weight and feel less depressed. Among obese participants, who feel more helpless regarding their weight, the factor that affects their depression is often the degree of stigmitization and how sensitive they are to it. Younger adolescents are at a more sensitive age to these issues and are thus more deprssed when obese. These findings point to the importance of identetification and treatment of depression among Palestinian children/adolescents attending treatment for over-weight or obesity. The study clearly showed that among over-weight and obese Palestinian children and adolescents (9–16 year old) experience more depression symptoms than those with normal weight. As such, it is critical to offer psychological treatment and emotional support as part of any weight loss intervention program for youth. Especially as these adolescents' families might encourage them to avoid seeking professional help and deal with the problem in the family, it is very important to identify them early, before the problem escales. Identification of those at risk to develop depression and providing early intervention, might save these youth from severe psychological disorders and even death from suicide later on, if their depression goes untreated.

**Open peer review.** To view the open peer review materials for this article, please visit http://doi.org/10.1017/gmh.2024.126.

**Data availability statement.** The datasets used and/or analyzed during the current study available from the corresponding author on reasonable request.

**Author contribution.** All authors contributed equally to this article, Hadeel Agbaria prepared the theoretical background section, Fayez Mahamid prepared methodology and analysis sections. Finally, Dana Bdier prepared the discussion section.

**Financial support.** No funding was received for this study.

**Competing interest.** The authors declare that they have no conflict of interest. All authors agreed in submitting the manuscript to the journal.

**Ethics approval and consent to participate.** All procedures performed in this study involving human participants were in accordance with the ethical standards of An-Najah National University IRB, the American Psychological Association, and with the Helsinki Declaration. Informed consent was obtained from parents all participants. The protocol of our study was received ethical approval from An-Najah National University IRB before data collection was initiated.

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
