## [Reviewer Report]

Please ensure that the references in the text are according to APA formatting

Please ensure that the references are in the right order

Please ensure that all the references in the text and referenced in the back of the references list

Please review the key words and consider revising.

Please ensure that the article is checked for structure and formatting.

---

## [Reviewer Report]

Dear Editor-in-Chief,

I would like to express my sincere thanks for the opportunity to review the manuscript titled “Differences in Severity of Depression Symptoms in Overweight, Obese, and Normal Weight Palestinian Children and Adolescents.” This study addresses a critical and timely issue, providing valuable insights into the mental health challenges faced by children and adolescents in a unique sociopolitical context. The research is particularly significant given the limited data available on this topic within non-Western societies.

I also wish to extend my gratitude to the authors for their work on this important subject.

Please find below some suggestions that may help further strengthen the manuscript:

1. Abstract: Including a brief mention of the study’s methodology would be beneficial. This addition would give readers a clearer understanding of the research approach.

2. Demographic and Weight Questionnaire (DWQ) (Page 9): A more detailed explanation of the DWQ’s purpose and relevance within the study would be helpful.

3. Discussion: The manuscript appropriately emphasizes the importance of conducting this study within a collectivist society that experiences political violence, contrasting it with the majority of studies that focus on Western societies. Expanding the discussion to further reflect the Palestinian context would be highly beneficial.

Thank you for considering these suggestions. I look forward to seeing the revised manuscript.

---

## [Reviewer Report]

The manuscript titled Differences in Severity of Depression Symptoms in Overweight, Obese, and Normal Weight Palestinian Children and Adolescents explores the relationship between weight categories and depressive symptoms among Palestinian children and adolescents aged 9-16. The study, involving 270 participants, highlights that overweight and obese children exhibit significantly higher depressive symptoms across several domains compared to their normal-weight counterparts. The study further examines the role of gender and age, revealing that overweight and obese girls generally show more severe depressive symptoms, while older adolescents in the overweight group experience worse symptoms than younger ones.

The manuscript is very interesting and has the potential to contribute significantly to the literature.

Areas for Improvement:

Clarification of Statistical Methods: Although the study uses ANOVA and post-hoc Tukey tests, more clarity on why these specific methods were chosen could enhance the manuscript’s rigor. Explaining how these tests control for potential confounders (like socioeconomic status) would strengthen the argument.

Cultural Considerations: The manuscript touches on the collectivist culture of the Palestinian context. However, further exploration of how cultural factors influence body image and weight-related stigma could provide deeper insights into the findings.

Effect Sizes: Including effect sizes (such as Cohen’s d or partial eta-squared) for the differences between weight groups and depressive symptoms would provide readers with a better understanding of the practical significance of the findings, not just the statistical significance.

Validation of Measurement Tools: The manuscript notes that the depression scale used was translated into Arabic for the first time. More details about the validation process and reliability of the translated tool would address potential concerns about measurement accuracy.

Incorporating these suggestions could improve the overall clarity, depth, and impact of the study.

---

## [Reviewer Report]

The reviewers have addressed the comments adequately, and I have no further comments at this time.